

# Leveraging the satellite-based climate data record CLARA-A3 to understand trends and climate regimes relevant for solar energy applications over Europe

Abhay Devasthale[1, *], Sandra Andersson[2], Erik Engström[2], Frank Kaspar[3], Jörg Trentmann[3], Anke Duguay-Tetzlaff[4], Jan Fokke Meirink[5], Erik Kjellström[6, 7], Tomas Landelius[1], Manu Anna Thomas[1], and Karl-Göran Karlsson[1]

[1]Meteorological Research Unit, Swedish Meteorological and Hydrological Institute (SMHI), Folkborgvägen 17, 60176 Norrköping, Sweden
[2]Community Planning Service, SMHI, Folkborgvägen 17, 60176 Norrköping, Sweden
[3]Deutscher Wetterdienst (DWD), Frankfurter Str. 135, 63067, Offenbach, Germany
[4]Federal Office of Meteorology and Climatology (MeteoSwiss), 8058 Zürich, Switzerland
[5]R&D Satellite Observations, Royal Netherlands Meteorological Institute (KNMI), 3731 GA De Bilt, The Netherlands
[6]Rossby Centre, SMHI, SE-601 76 Norrköping, Sweden
[7]Bolin Center for Climate Research, Stockholm University, SE-106 91 Stockholm, Sweden

*Correspondence to*: Abhay Devasthale (abhay.devasthale@smhi.se)

**Abstract.** Efficient transitioning to renewable energy requires fundamental understanding of the past and future climate change. This is particularly true in the case of solar energy since the surface incoming solar radiation (SIS) is regulated heavily by atmospheric essential climate variables (ECVs) such as aerosols and clouds, and by their long-term trends. Given the
complexity of the interactions and feedbacks in the Earth system, even small changes in ECVs could have large direct and indirect effects on SIS. The net efficacy of the designed solar energy systems therefore depends on how well we account for the role of ECVs in modulating SIS at decadal scales. In this study, by leveraging the satellite-based climate data record CLARA-A3, we investigate the recent trends in SIS and cloud properties over Europe during the 1982-2020 period. Further, we derive emerging climate regimes that are relevant for solar energy applications. Results show a large-scale increase in SIS
in spring and early summer over Europe, particularly noticeable in April and June. The corresponding trends in cloud fraction and cloud optical thickness, and their correlation with SIS suggest an increasingly important role of clouds in defining the favorable and unfavorable climate regimes for solar energy applications. We note also a strong spatio-temporal variability in trends and correlations. The results provide valuable metrics for the evaluation of climate models that have a dynamically integrated solar energy component.





## 1. Introduction

The share of renewable energy sources in the European Union has increased from 12.5% in 2010 to 23% in 2022 following the Renewable Energy Directive (2009/28/EC). The latest binding renewable energy target will increase that share to at least 42.5% by 2030 (https://energy.ec.europa.eu/topics/renewable-energy/renewable-energy-directive-targets-and-rules/renewable-energy-directive_en). The European Green Deal is paving the way for a faster transition towards cleaner energy. Transitioning to solar energy is happening at an even faster rate with many EU member states projected to reach their 2030 targets well ahead in time according to the latest, revised National Energy and Climate Plans (NECP) (https://energy.ec.europa.eu/topics/energy-strategy/national-energy-and-climate-plans-necps_en).

To facilitate more efficient transitioning to clean renewable energy, a better understanding of the past and future climate change is required (Jerez et al., 2015; Engeland et al., 2017; Grams et al., 2017; Gernaat et al., 2021; Hou et al., 2021; Dutta et al., 2022; Dong et al., 2023; Ha et al., 2023; Kapica et al., 2024). All three major sources of renewable energy (i.e. hydro, solar and wind) are subjected to the influences from the changing essential climate variables (ECVs) such as surface incoming solar radiation (SIS), precipitation, winds, temperature and humidity to name a few. If we were to design solar energy systems that are highly efficient and reliable also in the near future, a detailed understanding of both past and future changes in incoming solar radiation at the surface and the drivers behind its spatio-temporal variability is of paramount importance.

Thanks to forty years of near-continuous and global observations from the combined meteorological satellites of the U.S. (NOAA) and Europe (MetOp), it is now possible to derive valuable, long-term information on cloud properties and surface solar radiation (Cano et al., 1986; Pfeifroth et al., 2018ab; Devasthale et al., 2022; Devasthale and Karlsson, 2023; Karlsson et al., 2023). Recent studies have demonstrated and argued the importance of satellite-based observations to support the transitioning to renewables in general (Kaspar et al., 2019; Druecke et al., 2021; Edwards et al., 2022) and to solar energy in particular (Campana et al., 2020; Darragh and Fiedler, 2022; Hammer et al., 2023). Importantly, there have also been significant improvements in the calibration and retrieval algorithms in the recent decades,





uplifting them to climate quality. This enables the derivation of more stable and mature climate data records (CDRs) of various ECVs that are increasingly suitable for climate change studies.


In light of the aspects mentioned above, the holistic purpose of the present study is to demonstrate how we can exploit the satellite-based CDRs to distil and convey information on surface solar radiation to help facilitate transitioning to solar energy. In practice, we aim to answer the following three specific questions.

a) Can we derive user-friendly information on climate regimes of relevance for solar energy applications over Europe?

- This is a completely novel value addition. Here, we attempt to combine trends in surface solar radiation with trends in cloud properties and meteorological variables to distill useful information on spatio-temporal features in emerging climate regimes potentially favourable or unfavourable for solar energy

applications.

b) How well do cloud properties correlate with SIS?

- This is also a novel aspect of the present study considering the spatio-temporal scales. Here, the aim is to assess the role of clouds as one of the main drivers of the spatio-temporal variability in SIS.

c) What are the recent trends in SIS and cloud properties?

- This is a complementary assessment to previous studies in order to further deepen the understanding of spatio-temporal trends in SIS and cloud properties. The exact value addition of the present study will be to assess very detailed trends at the monthly scales, and use the most recent and longer-term information from a polar orbiting satellite-based CDR.

**2. Satellite-based cloud and radiation climate data record**

The third edition of the EUMETSAT's Satellite Application Facility for Climate Monitoring (CM SAF) cLoud, Albedo and surface RAdiation dataset from Advanced Very High Resolution Radiometer (AVHRR) data, CLARA-A3, provides the retrievals of incoming solar radiation at the surface (SIS), cloud fraction, and cloud physical properties (Karlsson et al., 2023). The consistent retrievals of these

variables provide a unique opportunity to use them in the context of solar energy applications. CLARA-



A3 has a long history of dedicated and continuous developments dating back to its beginning 25 years ago in the framework of EUMETSAT's Satellite Application Facility on Climate Monitoring (CM SAF). Furthermore, CLARA-A3 offers substantial improvements to its previous version, CLARA-A2 (Karlsson et al., 2017). A number of previous studies have documented the theoretical basis, validations and improvements in the CLARA-A3 climate data record.

In this specific study, we use the Level 3 monthly means of cloud and radiation products that are available at a 0.25 degree spatial resolution globally. The AVPOS version (i.e. AVHRRs onboard polar orbiting satellites) of this dataset is analysed here (https://www.cmsaf.eu/EN/Products/NamingConvention/Naming_Convention_node.html). The AVPOS version refers to the fact that the Level 3 data are prepared using quality-controlled retrievals from AVHRR sensors flying onboard all available polar orbiting NOAA and MetOp satellites, instead of using only one prime morning or afternoon NOAA and MetOp satellite at a time. The CLARA-A3 CDR currently covers the period from 1979 to 2020 with an Interim CDR thereafter. In this study, we use data from 1982 through 2020. The earliest data between 1979 and 1981 from TIROS-N and NOAA-6 show spurious behavior and are deemed not suitable for trend analyses. We analyse SIS, daytime cloud fraction and cloud optical thickness of liquid and ice clouds. We are thus leveraging the CLARA-A3 CDR by making maximum use of the valuable information on climate variables provided in this CDR.

## 2.1 Cloud property retrievals

CLARA-A3 contains a wide range of cloud properties. Cloud detection is based on Naïve Bayesian theory, employing global matchups between AVHRR and CALIPSO-CALIOP data for training. The algorithm yields a cloud probability that is reduced to a binary cloud mask (using a 50% probability threshold) for downstream retrievals. Cloud top height, pressure and temperature are derived using an artificial neural network, likewise trained with collocations between AVHRR and CALIPSO-CALIOP. Cloud phase is determined with a series of spectral tests applied to the AVHRR infrared channels. Cloud optical thickness and particle effective radius are simultaneously retrieved during daytime using the classical Nakajima and King (1990) approach by fitting observed reflectances in a visible and shortwave-



infrared channel pair to pre-calculated look-up tables of top-of atmosphere reflectances for cloudy atmospheres. Further details can be found in Karlsson et al. (2023) and references therein.


## 2.2. Retrievals of incoming solar radiation at the surface (SIS)

The estimation of the surface irradiance is based the probabilistic cloud mask and the top-of-the-atmosphere reflected solar radiation flux, both derived as part of the CLARA-A3 retrieval scheme (Karlsson et al., 2023). The cloud mask is used to separate clear-sky pixels from those that contain clouds.

For clear-sky pixels the surface irradiance is derived using a clear-sky radiation transfer model (Mueller et al., 2009). For cloudy pixels a look-up-table approach is used, which relates the surface radiation to the derived reflected solar flux (Mueller et al., 2009). Auxiliary data for the atmospheric columns of water vapor and ozone as well the surface albedo are taken from ERA-5; monthly climatological aerosol information is used to account for direct aerosol effects. For the estimation of daily averages from the

instantaneous satellite observations the diurnal cycle of solar radiation is considered; the monthly averages are derived from the daily averages. More details can be found in Karlsson et al., 2023.

## 2.3. Evaluation of SIS over Sweden

As mentioned in Section 2.2 above, the retrievals of SIS are validated over a large number of stations

located in Europe which is the focus region of this study (Riihelä et al., 2015; Urraca et al., 2017; Babar et al., 2019; Devasthale et al., 2022). Since we have in-house access to the SIS data from additional in-situ measurements at SMHI, we carried out further evaluation of SIS retrievals over these Swedish stations. We evaluated not only the latest CLARA-A3 SIS CDR, but also compared it with its previous edition CLARA-A2 to highlight recent changes. The location of those Swedish stations is shown in Figure

1. The details of the in-situ stations, their quality control and measurement principles can be found in (Carlund, 2011; Riihelä et al., 2015; Devasthale et al., 2022). It is worth pointing out that these stations cover a wide range of topographical and meteorological conditions over Sweden, ranging from very cold and dry, high mountain locations in the north, over coastal regions and inland areas, to the warm and wet regions in the southwest. This provides a good opportunity to evaluate CLARA-A3 SIS retrievals under

a range of surface and meteorological conditions.



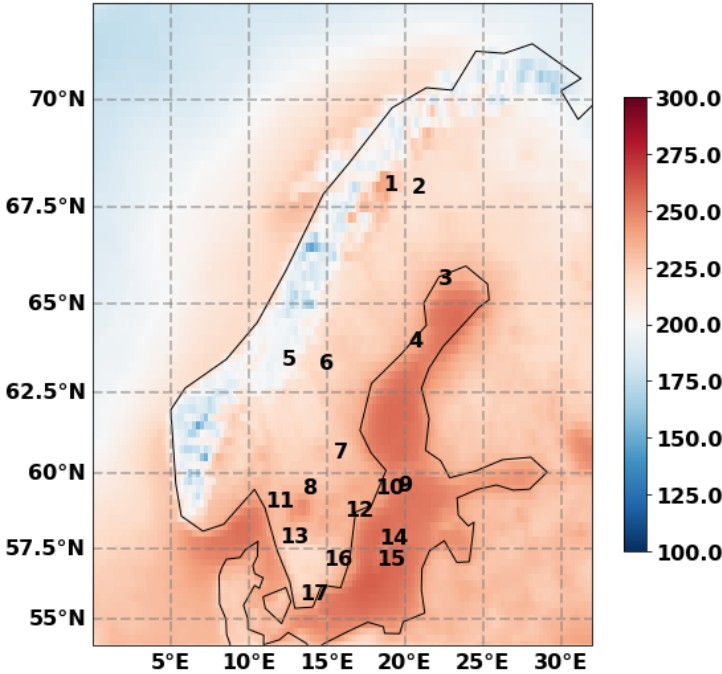

**Figure 1: The geographical positions of 17 Swedish SIS measurement stations used to evaluate SIS monthly means in the CLARA-A3 CDR. The background shows climatological mean SIS in W/m2 for the month of June (1991-2020) based on CLARA-A3.**


Figure 2 shows the results of SIS evaluations in terms of standard statistical metrics. It shows the Pearson correlation coefficient between the SIS measurements and CLARA-A2 and CLARA-A3 monthly means together with mean bias, standard deviation in bias and the root mean squared bias for the 17 stations that have long-term, quality-controlled SIS measurements. It is evident in Figure 2 that the correlations in

CLARA-A3 have improved in all but one station. The most noticeable improvements are seen in stations 1, 2 5 and 6. This is particularly encouraging, given the fact that the stations 1 and 2 are located in the high Swedish mountains, while the stations 5 and 6 are also located well in-land, but in the lower mountainous regions. The terrain around these four stations is very heterogenous with a mixture of snow-covered and bare mountains with the surrounding vegetation. The mean bias has also decreased in

CLARA-A3 for the majority of the stations. The stations 1, 2, 5, 6, 13 and 17 show significant reductions in root mean squared bias. These results indicate a clear improvement in the recent CLARA-A3 SIS CDR compared to its previous edition. Previous evaluations show that CLARA SIS tend to underestimate the





magnitude of trends compared to the in-situ measurements (Devasthale et al., 2022). Given the fact that we use only binary outcome of trends (i.e. increasing or decreasing) and that the total cloudiness in
CLARA-A3 satisfies the most stringent requirements of stability set by the WMO Global Climate Observing System (GCOS) over most of Europe (Devasthale and Karlsson, 2023), our mapping of climate regimes shown in Section 4 is expected to show robust spatial features.

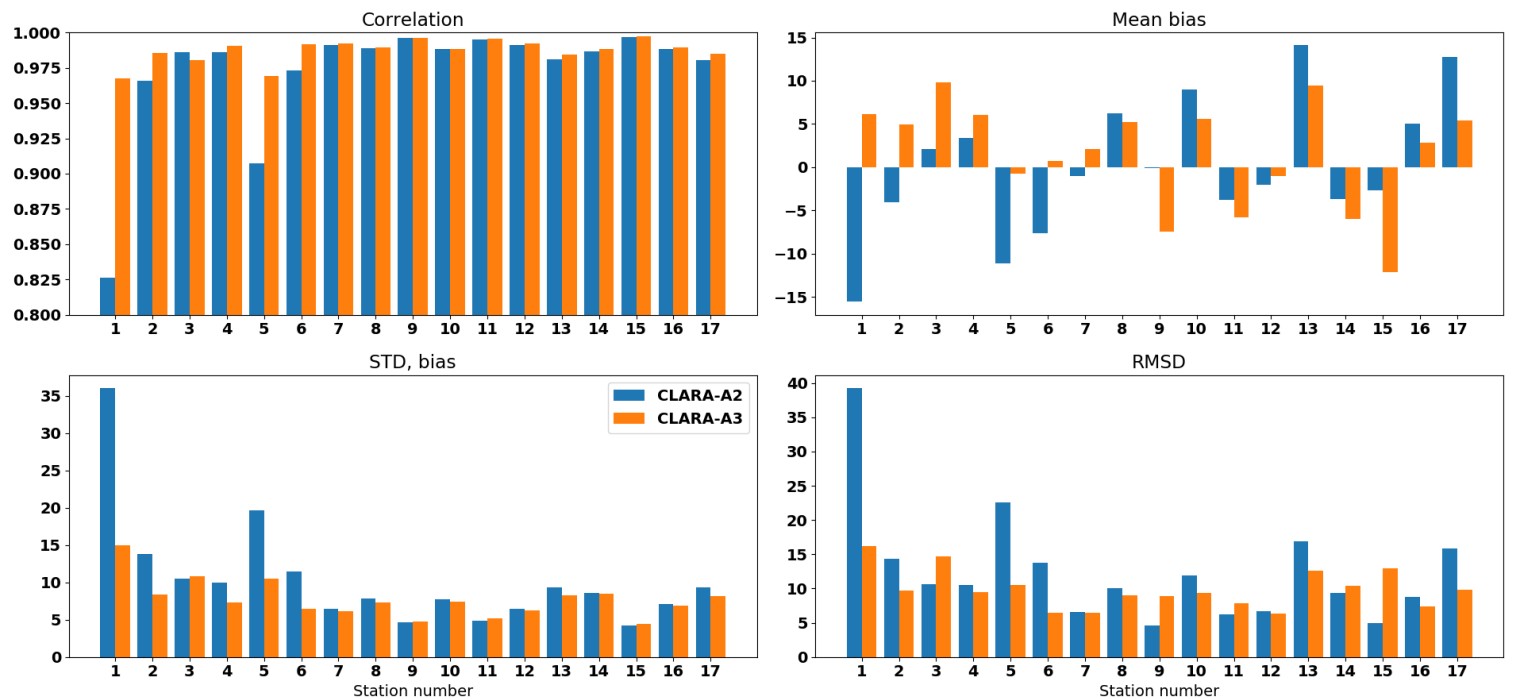


**Figure 2: Evaluation of CLARA-A2 and CLARA-A3 SIS against the Swedish in-situ measurements. The figures show correlation coefficients, mean bias, standard deviation of bias and root mean squared differences. The units are in W/m2.**

## 3. Trends in SIS and daytime cloud fraction

To assess whether regional climate regimes are subject to unfavourable developments for the suitability of solar energy production, we need to understand the absolute trends in the incoming solar radiation at the surface. Figure 3 shows the monthly trends in SIS over Europe based on the CLARA-A3 CDR covering the 1982-2020 period. To interpret these trends in SIS, Figure 4 shows the corresponding trends



in daytime cloud fraction. In January, the trends over most of Europe are either very weak or not

statistically significant. In February, an interesting east-west feature is observed, in that, eastern Europe

has experienced a decrease in SIS, while western Europe and the Mediterranean Sea have experienced an

increase in SIS. Although these trends are relatively weak, they are statistically significant. It is also to be

noted that SIS values are smaller in winter, so any trend in absolute terms would also be smaller. The

opposite trends in the eastern and western parts of Europe are also observed in daytime cloud fraction and

they anti-correlate strongly with the trends in SIS.

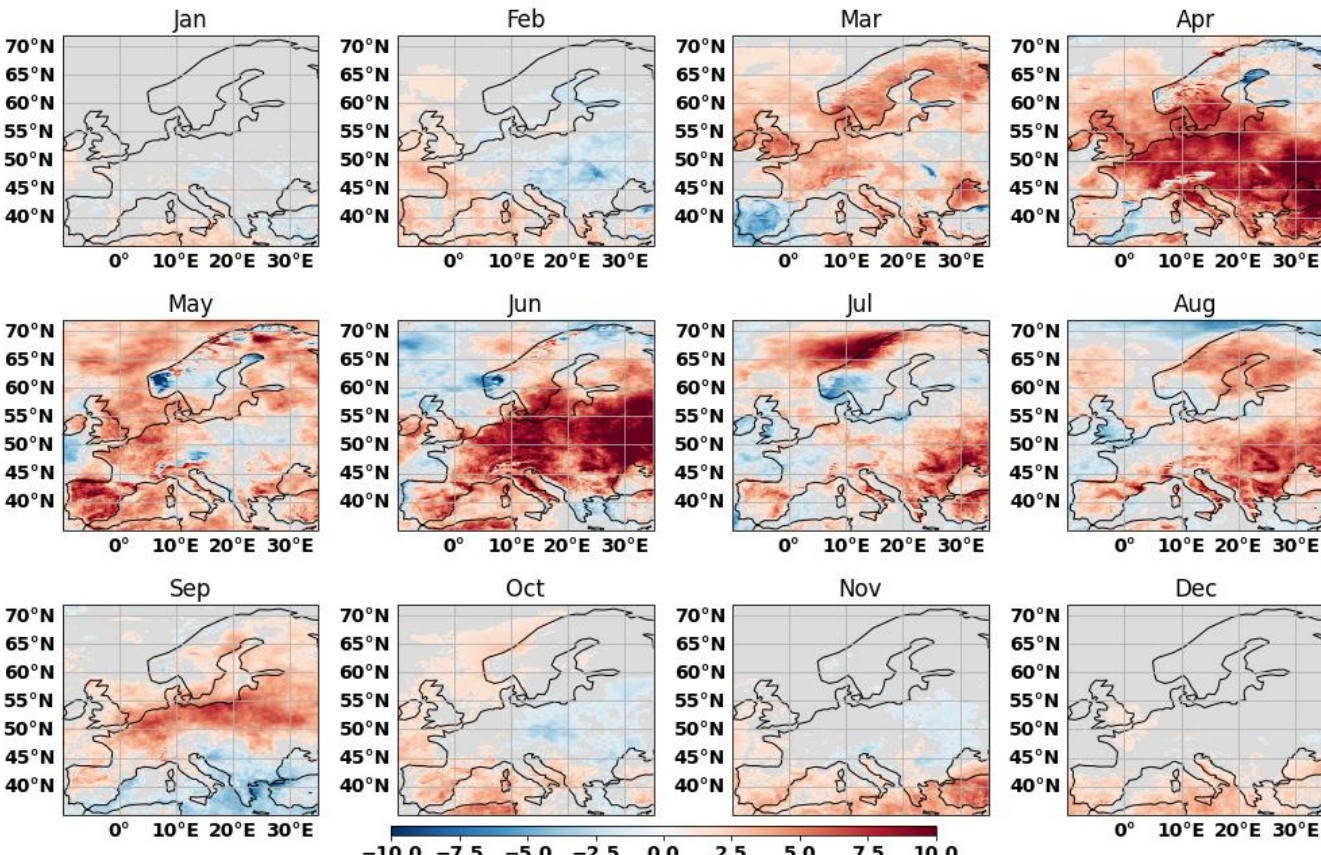

**Figure 3: The spatial trends in SIS (in W/m2/decade) based on the CLARA-A3 CDR (1982-2020). Only those trends that are significant at 90% confidence interval are shown.**

In March, except Spain, Portugal, the northern coast of Norway and few parts in east Europe, Russia, all

other European regions show increasing trends in SIS and decreasing daytime cloudiness. April shows



very strong increases in SIS and decreases in daytime cloudiness over most of Europe. The strongest trends are observed over southern Scandinavia and parts of central and eastern Europe. The daytime cloud fraction over these areas has decreased by almost 5% per decade in April, leading to a strong increase in SIS. The changes in cloudiness and SIS over northern Fennoscandia and the Iberian Peninsula are weaker, in some areas with reversed sign. The notable strong change in the European weather regime in April was also discussed by Ionita et al. (2020) and Imbery et al. (2020).

**Figure 4: The spatial trends in daytime cloud fraction (in %/decade) based on the CLARA-A3 CDR (1982-2020).**



In May, the Iberian Peninsula shows a strong increase in SIS of more than 5 W/m2/decade connected to a decrease in daytime cloudiness, especially along the entire northern and southeastern coasts. Also, other parts of western Europe, such as Germany, France, the Netherlands, and the United Kingdom, show increases in SIS. Over southern Scandinavia and eastern parts of Europe, the trends are not statistically significant.

June shows the most striking increases in SIS over central and eastern Europe reaching more than 6 W/m2/decade over southern Scandinavia, Germany, Switzerland, Italy, Belarus, Ukraine and western Russia. Large-scale decreases in daytime cloudiness are observed over the entire European continent in June. In July and August, the surface solar radiation has increased in southern regions of continental Europe. A very strong increase in SIS along the northern Norwegian coast and over the Norwegian Sea

in July and a corresponding decrease in cloudiness is also noteworthy.

In September, northern continental Europe shows increasing trends in SIS, especially in the latitude band 48N-55N. The central parts of Fennoscandia also show slight increase in SIS. The regions over and around the Mediterranean Sea show small, but statistically significant decrease in SIS in September. This is in slight contrast to October and November, when the Mediterranean regions show a small increase in SIS,

while northern European regions show small decrease or statistically insignificant changes. The changes in December are generally insignificant over much of Europe. It is to be noted that the spatial trends in daytime cloud fractions in November, December and January are very heterogeneous and have larger uncertainties compared to the other sunlit months of the year due to limited sampling in these winter months as the solar zenith angles are high.


## 4. Climate regimes relevant for solar energy

We begin by explaining the rationale behind why and how we derive the climate regimes relevant for solar energy applications. The shortwave solar radiation reaching the surface is regulated by a number of atmospheric components and their feedbacks in the backdrop of increasing greenhouse gases. Among

them, the most important are clouds and aerosols, which exert enormous influence on the spatio-temporal variability of SIS, and to some extent ozone. The changes in total daytime cloudiness is already discussed in Section 3 above. In Europe over the land regions, the primary sources of aerosols are anthropogenic.



The large policy changes in the late 1980s and the early 1990s in Europe have led to decreases in aerosol precursor gases (Vestreng et al., 2007) and particulate matter over the last decades, as pointed out in many previous studies (Cherian and Quaas, 2020; Yang et al., 2021; Glantz et al., 2022; Quaas et al., 2022). As a result, the brightening trends have been pointed out in a number of pioneering studies by Wild et al. (2005, 2009, 2021). Going forward in future, apart from the southern and eastern regions in Europe (Gutiérrez et al., 2020), that are episodically affected by desert dust outbreaks and biomass burning, the aerosols are not expected to play a seminal role in regulating SIS, especially in the northern parts of Europe where SIS has much stronger seasonality (Drugé et al., 2021).

The changes in ozone paint a complex picture. After recovering for few decades following the international agreements to reduce ozone depleting substances, the stratospheric ozone has again shown a slight decrease in recent years (Bognar et al., 2022; Villamayor et al., 2023). The research is currently ongoing to understand the drivers of this decrease, for example, the role of atmospheric dynamics and transport, the role of very short-lived halogens in the lower stratosphere and so on. Changes in stratospheric ozone over Europe are weak however not strong. The tropospheric ozone shows statistically insignificant changes over Europe in the last few decades, as the nitrogen oxide and its derivatives have also decreased in Europe (Yan et al., 2018; Zimke et al., 2019).

As aerosols and ozone are expected to have a relatively small role in future, changes in clouds and their opacity, are expected to exert much larger influence on SIS. Recent studies have already pointed out an increasing role of clouds in regulating the past trends in SIS over Europe and Scandinavia. Even small long-term changes in cloud cover and cloud optical thickness could have significant impacts on the efficacy of photovoltaics systems. An important question therefore is: How shall one capture the interplay among the trends in SIS, cloudiness and cloud opacity so as to better inform about the spatio-temporal nature of emerging climate regimes of relevance for solar energy applications?

In practice, this is illustrated here by compositing the combinations of the trends in SIS, daytime cloud fraction and optical thickness to highlight changes in the climate regimes. Each variable in question here (i.e. SIS, cloud cover and cloud optical thickness) can have either a decreasing or an increasing trend. By combining these two possible trend outcomes of three variables, the composites of a number of combinations can be made for each grid point. We refer to each such composite as a climate regime. Thus,





these regimes, that are essentially based on the interplay of trends in SIS, cloudiness and optical thickness, would be very relevant for solar energy applications. For example, a climate regime wherein SIS is increasing and cloudiness and cloud opacity are decreasing is certainly very favourable for solar energy applications compared to a contrasting climate regime wherein SIS is decreasing and cloudiness and cloud

opacity are increasing.

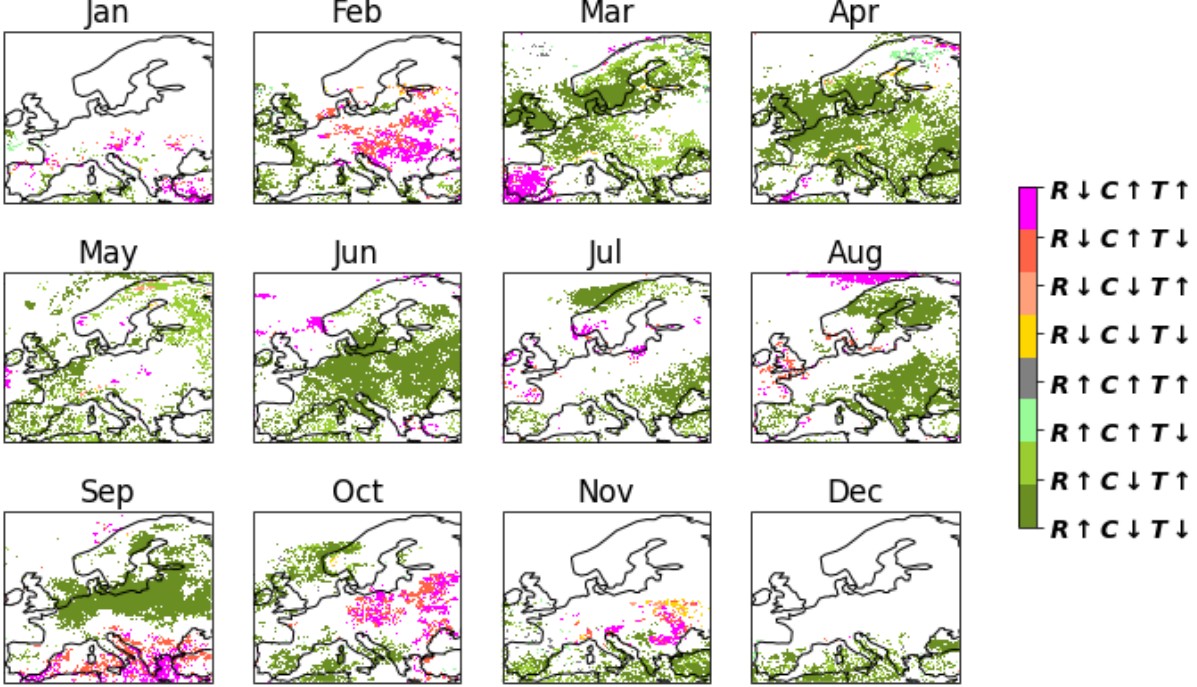

**Figure 5: The climate regimes based on the combination of trends in surface solar radiation (R), daytime cloud fraction (C) and liquid cloud optical thickness (T). The arrows show either increasing or decreasing trend in these variables. The white areas show the regions with either missing data or where the trends in either of**

**them are not statistically significant.**

Figure 5 shows these various climate regimes for each month. Here, together with the trends in SIS and daytime cloudiness, the trends in the in-cloud optical thickness of liquid phase clouds are considered. Figure 6 further shows the same, but when the trends in the optical thickness of ice phase clouds are considered. Instead of using the total optical thickness, we choose to show the results separately for liquid

and ice optical thicknesses due to the fact that the physical drivers and cloud controlling factors for low level liquid and high level ice clouds can be different. For example, the surface fluxes and boundary layer





have large impact on low level clouds, especially in the summer months, while the large-scale dynamics influences high ice clouds more strongly. While interpreting the climate regimes in Figures 5 and 6, the trends in SIS and its co-variability with cloudiness and cloud optical thickness need to be taken into

account. While the trends are already discussed in Figures 3 and 4, Figures 7-9 below show the correlations.

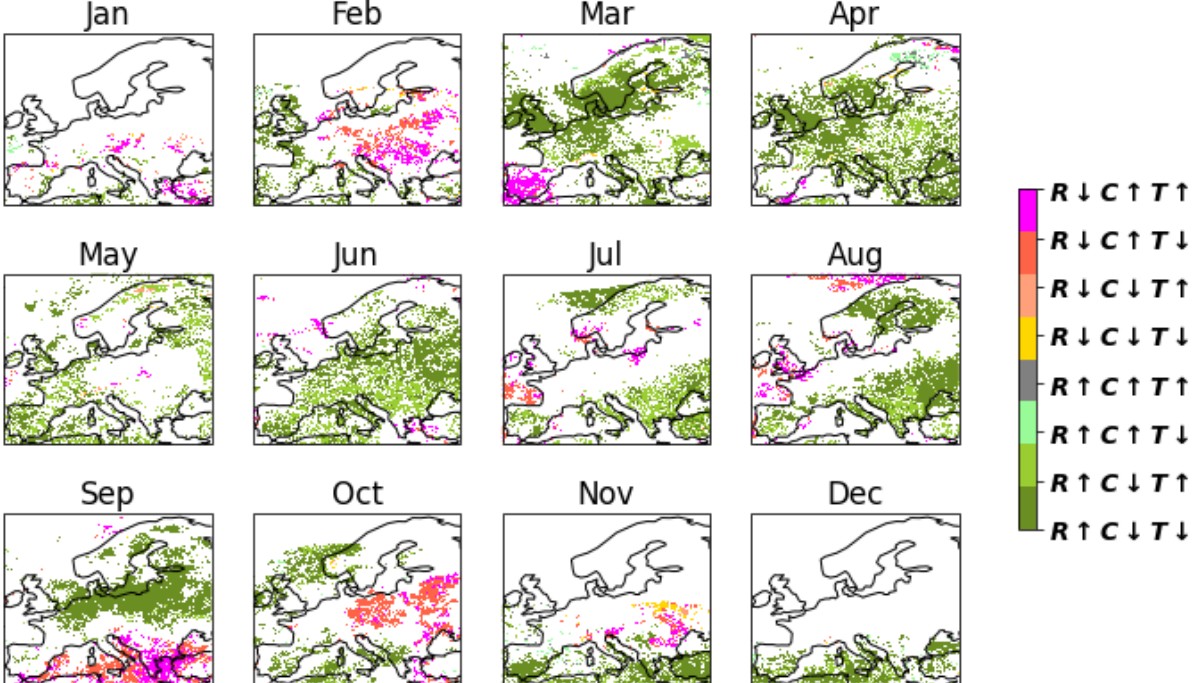

**Figure 6: Same as in Figure 5, but when ice cloud optical thickness is considered while deriving the climate regimes.**


The greenest colour in Figures 5 and 6  depicts the climate regime that is most favourable for solar energy applications. It shows the regions where the surface radiation is increasing (R↑), while simultaneously the cloudiness (C↓) and the optical thickness (T↓) are also decreasing. The existence of this regime is strongly evident in the months of April and June covering nearly entire Europe. The trends in SIS, cloudiness and

the correlation among them is very strong in these months. This regime also dominates the parts of central and eastern Europe and Scandinavia in March and September, as expected from Figures 3-4 and Figures 7-9. The next lighter shade of green also shows a favourable climate regime, wherein the surface radiation



in increasing (R↑), cloudiness is decreasing (C↓), but the optical thickness is increasing (T↑). The cloud fraction and cloud optical thickness can be independently influenced by a number of factors, such as temperature, humidity, aerosol composition, size and number density, and atmospheric dynamics. In a warming world, the water holding capacity of air also increases as the temperatures increase. Even though the cloudiness may decrease, the optical depth of clouds can have increasing trends under certain conditions. The existence of R↑C↓T↑ regime is visible in the central and eastern parts of Europe in early spring in March and April in Figure 5 when the liquid and ice cloud optical thickness is considered and in southern Europe in June in Figure 6 when the ice phase optical thickness is considered. It is to be noted that over almost all regions where the favourable R↑C↓T↓ and R↑C↓T↑ regimes are seen, the surface radiation is negatively correlated with the liquid cloud optical thickness more strongly than with the ice cloud optical thickness (Figures 8-9). This means that the cloud thermodynamic phase also plays an important role in regulating the surface radiation.

The potentially unfavourable climate regimes are depicted by the red shades in Figures 5 and 6. In these regimes, surface radiation decreases (R↓) and cloudiness simultaneously increases (C↑). In case of the most unfavourable climate regime, the cloud optical thickness is also increasing (T↑). In this case, the atmospheric interference with the incoming solar radiation is strongest, mediated mainly through the changes in cloud properties. The existence of unfavourable regimes can be seen in February in parts of central and eastern Europe, in March in the Iberian Peninsula, in September in the Mediterranean region, and in October in parts of eastern Europe. The presence of unfavourable regimes is also seen over the United Kingdom in August.

There are further interesting features in Figures 7-9 that are worth noting. In general, the correlation of SIS is strongest with the daytime cloud fraction, followed by with the liquid and ice cloud optical thicknesses respectively. There is also a strong seasonal and spatial character to these correlations. For example, the correlations are stronger during the summer half year and they are spatially very heterogeneous. The results presented in Figures 3-9 show the complexity of interactions between clouds and surface radiation, but nonetheless point out the increasing importance of cloud properties compared to aerosols in regulating the surface radiation over Europe.






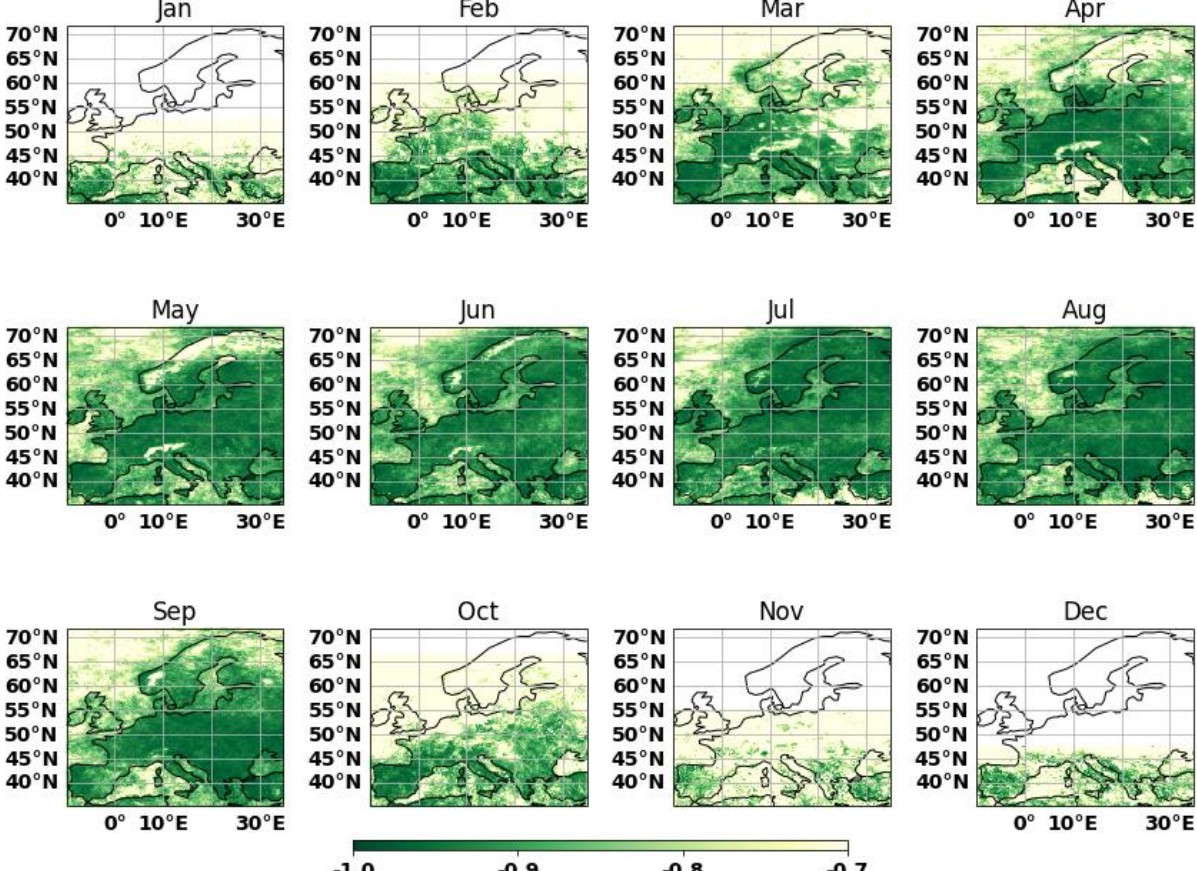

**Figure 7:** The correlation between SIS and daytime cloud fraction. The white areas show regions where the daytime cloud fraction data are not available.







**Figure 8: The correlation between SIS and liquid cloud optical thickness.**







**Figure 9: The correlation between SIS and ice cloud optical thickness.**

## 5. The meteorological context

In addition to the atmospheric constituents such as clouds, aerosols and ozone, the meteorological context also plays an important role in determining efficiency of the solar energy systems. For example, both temperature and atmospheric moisture influence the net performance of the systems. Figure 10 therefore shows the interplay among trends in surface solar radiation (R), total column water vapour (H) and surface temperature (T). The trends in total column water vapour and surface temperature were derived using the

ERA5 reanalysis data (Hersbach et al., 2020).



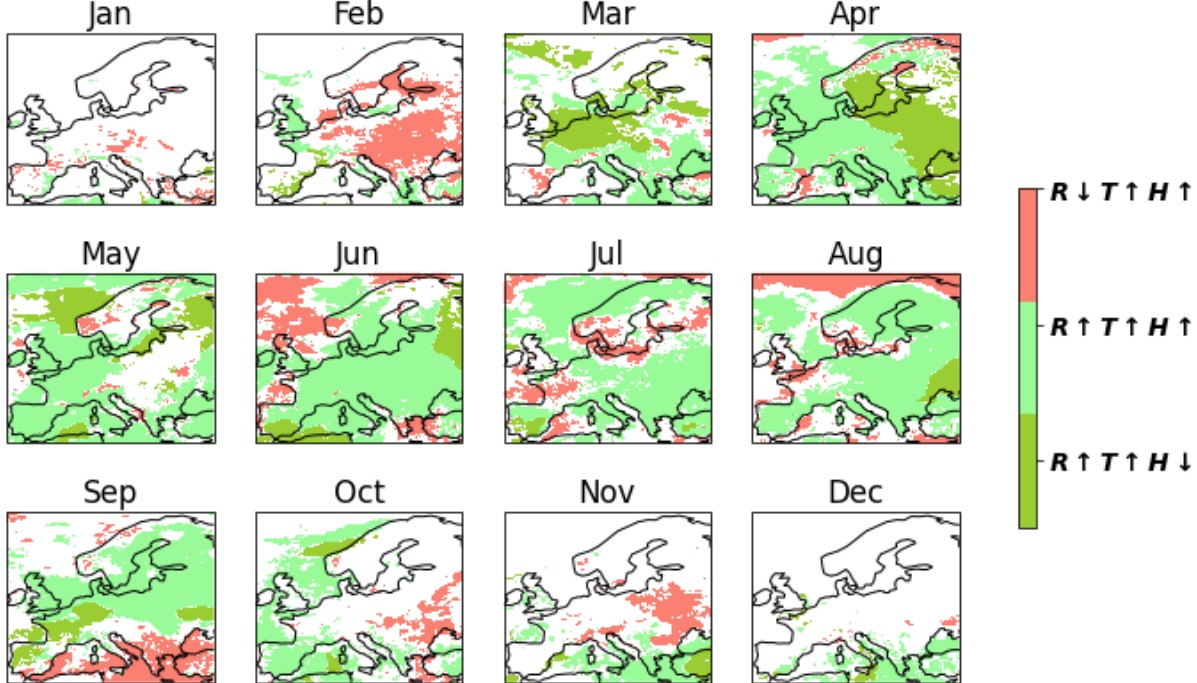

**Figure 10: The interplay among trends in surface solar radiation (R), surface temperature (T) and total column water vapour (H). The arrows show either increasing or decreasing trend in these variables. The white areas show the regions where the trend in either of them is not statistically significant.**

This interplay is dominated by three regimes. The first regime where both R and T are increasing, while H is decreasing (shown by the olive green colour). This is probably the most favourable meteorological regime for solar energy applications since all three variables in question have trends that would help increase the net performance of the solar energy systems from early spring to late autumn. In the second regime, all three variables show increasing trends (shown by the light green colour). Although the total column water vaour is increasing and will interfere with the incoming solar radiation, given the fact that the R and T are still increasing, this regime could also be considered as favourable. The third regime dominating the interplay, shown by the orange colour, is probably the least favourable, wherein the incoming solar radiation is decreasing.

The presence of R↑T↑H↓ regime over the northern central Europe in March and over the southern Scandinavia and eastern parts of Europe in April is noteworthy, indicating the emergence of favourable





conditions early in the spring. The month of June stands out as the month when the R↑T↑H↑ regime is dominating over the entire Europe. The third regime R↓T↑H↑ is dominating in February in the eastern and central Europe as well in the late autumn in October and November months in the parts of eastern

Europe. The parts of southern Scandinavia in July and August and the United Kingdom in August show decreasing trends in the surface solar radiation as can also be seen in Figure 3. The trends in surface temperature and total column water vapour are not always agreeing with one another and are spatially heterogeneous. This interplay shown in Figure 10 is broadly consistent with the climate regimes discussed in Figures 5 and 6 in the sense that the geographical distribution of favourable regimes is similar in both

cases, thus strengthening their applicability.

## 6. Discussions and conclusion

Knowledge about ongoing climate change is important as we transition to renewable and strongly

weather-dependent energy sources. A key aspect going forward is how to best improve this knowledge by utilizing the state-of-the-art observations. In this context, we set out to answer the following three questions.

a) Can we derive information on climate regimes that are of relevance for solar energy applications over Europe that is user-friendly and helpful to decision makers?

- We demonstrated that it is certainly possible to leverage modern climate data records to derive information that could be useful to decision and policy makers. The derivation of various climate regimes should help in the assessment of the state of climate relevant for solar energy applications. The existence of favourable climate regimes over Europe shows promise in increasing the exploitation of solar energy during the spring and early summer months. It is clear that the satellite-based CDRs can only describe the

recent past and not the near future. However, the recent state of the climate and changes therein often serve as predictor and basis for policy making in near future. This is one of the reasons the World Meteorological Organization (WMO) recommends computing climate normals (WMO, 2017; Devasthale et al., 2023). Given the urgency of energy transitioning in the coming few decades, the assessment of recent climate regimes, such as the one presented here, is even more relevant. These climate regimes





could furthermore be used as evaluation metrics to investigate the fidelity of climate models in capturing the drivers behind the trends in SIS and cloud properties. As a result, the climate models could be used more reliably to project the climate trends that are favourable for designing and implementing solar energy systems in the near future.

b) How well do cloud properties correlate with SIS?

- Our results showed very strong correlation of SIS with daytime cloud fraction, often exceeding 0.90 over large parts of Europe and during the summer half year. The correlations with the liquid cloud optical thickness are also very strong, while the correlations with the ice cloud optical thickness are relatively weaker. There is a strong spatio-temporal variability in these correlations.

c) What are the recent trends in SIS and cloud properties?

- The latest third edition of the CLARA CDR confirmed the large-scale increase in SIS over much of Europe during spring and early summer, complementing earlier studies. We further showed that this SIS increase is accompanied by large-scale decreases in daytime cloud fraction and cloud opacity. The increasing trends in SIS for April and June stand out together with remarkable decrease in cloudiness in those months over the last four decades. The outstanding change in the European weather regime in April was also discussed by Ionita et al. (2020) and Imbery et al. (2020).

All of the results presented above point to an increasing control of SIS by clouds. In future, this strong co-variability between SIS and cloud properties poses a number of challenges. Clouds are still notoriously difficult to represent in climate models (https://www.ipcc.ch/report/ar6/wg1/chapter/chapter-7). They are often pointed out as the largest source of uncertainties and the large spread in the equilibrium climate sensitivity is often attributed to our limited knowledge of future cloud feedbacks in the Earth system. This has direct implications for designing and implementing solar energy systems while using information of future cloud and radiation conditions from climate model projections. At the same time, it also implies that the satellite-based climate monitoring of co-variability between SIS and clouds would need to be strengthened even more to continue to provide a robust scientific basis for assessments that are of relevance for future solar energy applications.



**Code and data availability:**

All datasets used in the study are publicly available.

CLARA-A3 dataset can be accessed here:

https://wui.cmsaf.eu/safira/action/viewDoiDetails?acronym=CLARA_AVHRR_V003

SMHI station data can be accessed here:

https://www.smhi.se/data/meteorologi/stralning

ERA5 reanalysis data are obtained from the Copernicus Climate Data Store:

https://cds.climate.copernicus.eu/cdsapp#!/dataset/reanalysis-era5-single-levels-monthly-means

**Author contribution:** AD designed the study, carried out the analysis and wrote the first draft. All authors contributed equally to the interpretation of results and writing thereafter.

**Competing interests:** The authors declare that they have no conflict of interest.

**Funding:** This research was funded by Swedish Research Council (grant number 2021-05143) and the Swedish Government's 2023 Climate Adaptation Grant 1:10 to SMHI.

**Acknowledgements:** AD would like to thank the entire team of CM SAF/EUMETSAT.

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
