# Peer review of "Leveraging the satellite-based climate data record CLARA-A3 to understand the climatic trend regimes relevant for solar energy applications over Europe"

_EGUsphere, 2024_

## Author Comment (AC1)

**Response to Referee #1**

The authors present a research on solar radiation conditions over Europe, based on EUMETSAT's CMSAF CLARA-A3 satellite data record. Their analysis includes also influencing factors and trends, and they introduce a new concept called climate regimes, which may be helpful for a variety of applications including future planning of solar energy. Overall, the paper is interesting, well-written, and interesting for the readers of Egusphere.

We thank the referee for constructive comments and for the encouraging remarks. Please find below point by point reply to your comments.

In the abstract and elsewhere, the authors state the conclusion that their results suggest an increasingly important role of clouds for solar radiation conditions. I have difficulties understanding which part of the presented results would really show that this is true. From general knowledge about aerosol trends, climate change and so on, it would seem possible (or perhaps even plausible), but from the results presented in this paper, I do not see how they would confirm that clouds today are more strongly regulating solar radiation conditions than, for example, 20 years ago. I suggest revising and being careful about putting forward only such conclusions that are confirmed by the results of this paper.

We arrived at this conclusion based on the following results and facts. We see a clear decreasing trend in cloudiness over much of Europe as shown in Figure 4. At the same time, the aerosol load has also decreased in Europe over the same time period. Given that fact the trends in incoming surface radiation presented here are for all-sky conditions and the diminishing role of aerosols, the atmospheric opacity is then driven primarily by clouds in the recent decades. The relative change in cloud fraction is therefore having larger bearing on SIS than the relative change in aerosols. Few previous studies have also arrived at similar conclusions in different contexts (see e.g. Pfeifroth et al., 2018; Schillinger et al., 2024).

I also have some hesitation to accept the term climate regimes, which as used here more or less can be understood to be the composite of trends in SIS, cloud fraction, and cloud optical depth. Somehow, I find it a bit contradicting to call a trend composite climate regime. For me, it would be more natural to call it climate trend regime or something similar. Maybe the authors have already given this a lot of thought, but I would anyhow careful consideration once more on what term to use.

In the hindsight, we do agree as well. That term is probably misleading or confusing. This issue is raised by both referees. Therefore, the term "climate regime" is now replaced by "climatic trend regime" since we are indeed presenting the dominant, emerging composites of long-term trends in climate variables.

I recommend minor revisions.

**Specific comments and suggestions**

- L21-22 ("Net efficacy…"): a bit difficult to understand, perhaps a bit sloppy sentence. How would that knowledge influence the net efficacy?

  Here we are referring to the fact that, when not properly accounted for, any one of the modulators of incoming radiation (i.e. clouds, aerosols etc) could affect the efficiency and performance of solar energy system at a given location.

- L56: Hammer et al., 2023, in reference list there is only one Hammer et al., with the year 2003.

  Corrected.

- L71-72: I believe correlation between clouds and SIS is something that has been studied before. There is probably a wealth of publications based on ground-based measurements. Also, there is one recent satellite-based study (Post and Aun, 2024; https://doi.org/10.1016/j.oceano.2023.11.004) that is of particular interest here, worthwhile including in the references and discussing in conjunction with the results of the present paper. Post and Aun look at changes in cloudiness and SIS and how they correlate, and, in addition, they also evaluate the trends in the light of atmospheric circulation patterns.

  Thank you for pointing out this article. It is certainly relevant for our study and is cited in the revised version.

- Figure 1 (and other figures): I suggest adding a label and unit to the color bar, and a title to the plot, so that the figure is more easily accessible stand-alone

  Added.

- L148 (and elsewhere as needed): Oxford Languages (online resource) explains that bias is "a systematic distortion of a statistical result". This is also how I understand bias: it is a systematic feature, that is the aggregated result of individual errors in a set of data. Thus, it does not make sense in my mind to calculate the standard deviation of the bias, but the standard deviation should be calculated from the errors (or differences). Same goes for the root-mean-squared error.

  We agree. The text and labels are now changed to indicate that we are showing the differences.

- Figure 2 and L156: figure uses RMSD while text says root mean squared bias.

  The text is corrected.

- L191-192: Here, as a reader, I would like to have a bit more explained about what these studies found, without having the read them. Post & Aun (2024) mentioned above is relevant also here.

These studies are elaborated in the revised version.

- L221: shouldn't water vapour column also be mentioned here?

Yes, it is now mentioned.

- L222: Please provide a reference for this sentence

Cited the review by Wild, 2009.

- L236: Check sentence

Corrected.

- L240-241: Which studies? Please provide references.

Cited the references Pfeifroth et al., 2018; Devasthale et al., 2022; Post and Aun, 2024; Schillinger et al., 2024.

- L250 (and elsewhere throughout the paper): Definition of the term climate regime, as discussed already above.

This is now changed to climatic trend regime throughout the revised manuscript.

- L262: check sentence (in-cloud?)

"In-cloud" a standard terminology used to indicate that the retrievals of cloud optical thickness or water path are for cloudy pixels and are not normalized by any factor or grid averaged. The latter is usually used to compare satellite derived cloud properties with climate models.

- L342: How does the temperature and the moisture influence solar energy systems? References?

See the reply below.

- L352-354: Related to the comment above: How does increasing temperature help increase net performance of a solar energy system? It is known that for photovoltaics, increasing cell temperature decreases the relative efficiency of the system.

As mentioned by the referee, the higher temperatures actually reduce the efficiency of the PV systems. Increased humidity can also negatively affect the PV systems through dew formation and by causing physical degradation. In response to increasing greenhouse gases, we see increasing temperature trends in all months irrespective of the regime in question. So, the trends in humidity and SIS therefore decide which meteorological context could be favourable or unfavourable.

This is elaborated in the revised version and the new references of Dubey et al., 2013, Driesse et al., 2022, and Sengupta et al., 2024 are also cited.

- L356: spelling of water vapour. Also, maybe it would be helpful for readers to clarify whether the effect of water vapour is included in the CLARA SIS product (I guess it is).

  Corrected. Yes, the retrievals of SIS consider the atmospheric state, including the total integrated water vapour, based on the ERA5 reanalysis data. The details can be seen in the Algorithm Theoretical Basis Document here: https://www.cmsaf.eu/SharedDocs/Literatur/document/2023/saf_cm_dwd_atbd_clara_rad_3_3_pdf.pdf?__blob=publicationFile&v=2

  This is included in the revised text.

- L401: please provide references

  Provided.

---

## Author Comment (AC2)

**Response to Referee #2**

In this manuscript, an innovative approach is presented to analyze the long-term trend of SSI. The proposed concept is based on the synthesis of trends of SSI, cloud cover, cloud optical depth and humidity into a single quantity called "climate regime". The methodology is applied to satellite derived cloud properties and SSI from CLARA-A3 over Europe. The paper is very interesting, well-written and represents a very relevant contribution to readers of EGU sphere as well as the solar energy community to better quantify the effect of climate change on the solar resource.

I suggest the following minor revisions.

We thank the referee for appreciating our approach and for the encouraging remarks. Please find below point by point reply to your comments.

Line 173: "covering the 1982-2020 period » Please precise that according to recent works on SSI trends, a steady brightening is expected over this period.

Added.

The term "climate regimes" can be misleading in the context of the paper as it can lead to confusion with large-scale circulation regimes. I would suggest using a different term such as e.g. "multi-variable trend class".

In the hindsight, we do agree as well. It is probably misleading or confusing. This issue is raised by both referees. The term "climate regime" is now replaced by "climatic trend regime" since we are indeed presenting the dominant, emerging composites of long-term trends in climate variables.

Since aerosol and water vapor are affecting SSI, it would be important to clearly explain how these are treated in the retrieval of the cloud property (atmospheric correction) in section 2.1.

The retrievals of SIS consider the atmospheric state, including the total integrated water vapour, based on the ERA5 reanalysis data. The details can be seen in the Algorithm Theoretical Basis Document here: https://www.cmsaf.eu/SharedDocs/Literatur/document/2023/saf_cm_dwd_atbd_clara_rad_3_3_pdf.pdf?__blob=publicationFile&v=2
With regard to aerosols, the algorithms attempt to separate heavy aerosol loadings from clouds. Subsequently, in the cloudy scenes, aerosols are not considered. This is justified because aerosols are usually below the clouds and have a negligible optical depth compared to the clouds, although there are exceptions, e.g., absorbing aerosol above clouds, dust-infused clouds.

In lines 219-220, the authors wrote "The shortwave solar radiation reaching the surface is regulated by a number of atmospheric components and their feedbacks in the backdrop of increasing greenhouse gases. Among them, the most important are clouds and aerosols,… ». The absorption by water vapor is very important for SSI calculation but it is not mentioned in

this paragraph and not treated in section 4. In contrast, it is addressed in section 5. This is confusing and I think that reconsidering the organisation of these two sections would improve the flow of the text.

As mentioned in the reply above, the SIS retrievals consider the total column water vapour. The purpose behind Section 5 is to present the meteorological context, and to investigate if the climate trend regimes presented in Section 4 and the meteorological context presented in Section 5 show similar spatial coherence for the favourable conditions for exploiting solar energy.

SSI trends are assessed but it is not clear from the text if a statistical significance test has been applied. (it is briefly mentioned in L174-175). More details on this aspect would be helpful.

Yes, the statistical significance is applied to all trends presented in manuscript. This is mentioned in the figure captions.

Figure 5 and Figure 6 show the trend considering liquid and ice water COD respectively. It is unclear whether the dataset have been split or not for generating these two figures. This makes the maps not easy to interpret.

Both Figures 5 and 6 use daytime cloud fraction together with trends in liquid and ice cloud optical thickness separately. This is further clarified in the revised version.